# Influence of Bone Substitutes on Mesenchymal Stromal Cells in an Inflammatory Microenvironment

**DOI:** 10.3390/ijms24010438

**Published:** 2022-12-27

**Authors:** Siddharth Shanbhag, Neha Rana, Salwa Suliman, Shaza Bushra Idris, Kamal Mustafa, Andreas Stavropoulos

**Affiliations:** 1Center for Translational Oral Research (TOR), Department of Clinical Dentistry, Faculty of Medicine, University of Bergen, 5009 Bergen, Norway; 2Department of Immunology and Transfusion Medicine, Haukeland University Hospital, 5021 Bergen, Norway; 3Private Practice, Voss Tannspesialist Team, 5700 Voss, Norway; 4Division of Conservative Dentistry and Periodontology, University Clinic of Dentistry, Medical University of Vienna, 1090 Vienna, Austria; 5Department of Periodontology, Faculty of Odontology, Malmö University, 205 06 Malmö, Sweden

**Keywords:** mesenchymal stromal cells, bone substitutes, immune modulation, bone regeneration

## Abstract

Bone regeneration is driven by mesenchymal stromal cells (MSCs) via their interactions with immune cells, such as macrophages (MPs). Bone substitutes, e.g., bi-calcium phosphates (BCPs), are commonly used to treat bone defects. However, little research has focused on MSC responses to BCPs in the context of inflammation. The objective of this study was to investigate whether BCPs influence MSC responses and MSC–MP interactions, at the gene and protein levels, in an inflammatory microenvironment. In setup A, human bone marrow MSCs combined with two different BCP granules (BCP 60/40 or BCP 20/80) were cultured with or without cytokine stimulation (IL1β + TNFα) to mimic acute inflammation. In setup B, U937 cell-line-derived MPs were introduced via transwell cocultures to setup A. Monolayer MSCs with and without cytokine stimulation served as controls. After 72 h, the expressions of genes related to osteogenesis, healing, inflammation and remodeling were assessed in the MSCs via quantitative polymerase chain reactions. Additionally, MSC-secreted cytokines related to healing, inflammation and chemotaxis were assessed via multiplex immunoassays. Overall, the results indicate that, under both inflammatory and non-inflammatory conditions, the BCP granules significantly regulated the MSC gene expressions towards a pro-healing genotype but had relatively little effect on the MSC secretory profiles. In the presence of the MPs (coculture), the BCPs positively regulated both the gene expression and cytokine secretion of the MSCs. Overall, similar trends in MSC responses were observed with BCP 60/40 and BCP 20/80. In summary, within the limits of in vitro models, these findings suggest that the presence of BCP granules at a surgical site may not necessarily have a detrimental effect on MSC-mediated wound healing, even in the event of inflammation.

## 1. Introduction

Bone regeneration is the result of the interplay between osteogenic progenitor cells and immune cells [1]. Of these, mesenchymal stromal cells (MSCs), which give rise to osteoblasts, and peripheral blood monocytes, which give rise to macrophages/osteoclasts, are arguably the most relevant [2]. The carefully coordinated cellular and molecular events that drive wound-healing processes form the biological basis for the treatment of bone defects using guided bone regeneration or tissue engineering approaches [3].

MSCs are key players in the wound-healing process and have been shown to be the most active during the early stages of healing [4,5,6]. MSCs are hypothesized to promote bone regeneration via various mechanisms, including direct differentiation into osteoblasts, the paracrine stimulation of resident progenitor cells, the modulation of inflammatory and immune responses or a combination thereof [7], providing the basis for tissue engineering strategies for bone regeneration [8,9]. These processes, i.e., osteogenic differentiation, wound healing, immune modulation, remodeling, etc., are regulated by the interplay between several genes and proteins expressed by MSCs. For example, runt-related transcription factor 2 (*RUNX2*), a master transcription factor of osteoblast differentiation, and bone morphogenetic protein 2 (BMP2) together regulate MSC osteogenic differentiation [10,11]. Wound healing is mediated by growth factors, such as vascular endothelial growth factor (VEGF), basic fibroblast growth factor (FGF2), granulocyte colony-stimulating factor (GCSF) and platelet-derived growth factors (PDGF-AA/BB/AB). Interleukins (ILs) such as IL10 facilitate a “switch” from an inflammatory to a healing microenvironment, while others, such as IL5, IL6, IL7, IL8 and IL9, regulate inflammation and interactions with immune cells, e.g., macrophages (MPs), during the early stages of healing [12,13]. For example, MSCs have been shown to direct MP “polarization”, i.e., induce a phenotype shift from a pro-inflammatory (M1) phenotype towards an anti-inflammatory or pro-healing (M2) phenotype conducive to resolving inflammation and accelerating wound healing [14,15,16]. Moreover, in the later stages of healing, cytokines, such as the receptor activator of nuclear factor-κ β ligand (RANKL) and osteoprotegerin (OPG), regulate bone remodeling, especially in the presence of biomaterials [17].

Another factor usually present at an early bone healing site is the biomaterial; regardless of the clinical approach, most current strategies involve the use of bone substitutes for the treatment of alveolar and peri-implant bone defects [18,19]. Since human bone is composed of ~70% calcium phosphate (CaP), commonly used bone substitute materials are also CaP-based, e.g., hydroxyapatite (HA); β-tricalcium phosphate (β-TCP); or their mixtures, i.e., biphasic CaP (BCP). BCPs are commercially available as different products based on the ratio of HA/β-TCP, e.g., an HA/β-TCP ratio of 60/40 (BoneCeramic^®^, Institut Straumann AG, Basel, Switzerland) or an HA/β-TCP ratio of 20/80 (MBCP+^®^, Biomatlante, Vigneux de Bretagne, France). BCPs function as three-dimensional (3D) scaffolds for cellular attachment and growth (osteoconduction) during bone healing. Moreover, BCP-based bone substitutes are also used in tissue engineering strategies as scaffolds for MSC delivery [12,13]. In addition to functioning as scaffolds, bone substitutes may also provide instructive microenvironments to direct cellular (MSC) functions, such as differentiation, paracrine secretion and immune modulation [20,21]. Thus, the biomaterial may also modulate cellular responses and influence healing outcomes.

While several studies have investigated the cellular responses to bone substitutes in standard in vitro conditions [22], relatively little research has focused on the cellular responses to bone substitutes in the context of inflammation. From a clinical perspective, this is relevant since bone substitutes are often present at the sites of existing inflammation, e.g., peri-implant defects [23,24,25]. Moreover, in its early stages (48–72 h), the healing microenvironment is characterized by the presence of pro-inflammatory cytokines, primarily interleukin-1-beta (IL1β), tumor necrosis factor-alpha (TNFα) and interferon-gamma (IFNγ), which may further modulate cellular responses and regulate the healing process [26]. It is therefore of interest to study the effects of bone substitutes on cellular responses and interactions in an inflammatory microenvironment. Thus, the objective of the present study was to address the following research question: how do bone substitutes (BCP) influence MSC responses and MSC–MP interactions, at the gene and protein levels, in an inflammatory microenvironment? That is, does the presence of a bone substitute pose a risk for modulating early cellular responses and possibly delay healing at sites of inflammation?

## 2. Results

### 2.1. BCP Strongly Modulated MSC Gene Expression under Inflammatory Conditions

In setup A, the responses of primary human MSCs (Figure 1) to two different BCP granules (BCP 60/40 and BCP 20/80), with or without cytokine stimulation (IL1β + TNFα), were assessed via analyses of genes related to osteogenesis (*RUNX2* and *BMP2*), healing (*VEGF* and *IL10*), inflammation (IL6 and IL8) and remodeling (*RANKL* and *OPG*). The control cultures included monolayer MSCs with or without cytokine stimulation.

Overall, after 72 h, the presence of the BCP granules enhanced the expressions of the osteogenesis-, healing- and remodeling-related genes in the MSCs to varying degrees. In the presence of BCP 60/40, the expressions of the *BMP2, VEGF, IL10* and *RANKL* genes were enhanced compared to those of the control, while the inflammation-related genes (*IL6* and *IL8*) remained unchanged. Under inflammatory conditions, the BCP 60/40 granules further enhanced the expressions of these genes in the MSCs, in addition to *RUNX2, IL6* and *IL8* (Figure 2). The gene expression data were validated by performing an enzyme-linked immunosorbent assay (ELISA) for the BMP2 protein, which revealed a similar trend in levels under inflammatory and non-inflammatory conditions (Appendix A).

A similar trend in MSC gene expression was observed in the presence of the BCP 20/80 granules: *BMP2*, *VEGF*, *IL10* and *RANKL* were enhanced, in addition to OPG, compared to the control. In contrast to the BCP 60/40 culture, *IL6* expression was enhanced in the presence of BCP 20/80. Under inflammatory conditions, the BCP 20/80 granules further enhanced the expressions of *BMP2, IL10* and *RANKL* in the MSCs, along with those of the inflammation-related genes, i.e., *IL6* and *IL8* (Figure 3).

### 2.2. BCP Did Not Additionally Alter MSC Cytokine Profiles under Inflammatory Conditions

The secreted protein concentrations in the supernatant media of the BCP-cultured MSCs, with or without cytokine stimulation, were measured via a multiplex assay (setup A). Of the 27 tested cytokines, consistent and reliable readings were obtained for 11 cytokines related to healing (FGF2, VEGF, PDGF-BB, GCSF and IL10) and inflammation (IL5, IL7 and IL9). Additionally, chemokines such as C-C motif ligands 11 (CCL11), 4 (CCL4) and 5 (CCL5) were identified. The presence of the BCP granules, either 60/40 or 20/80, did not significantly alter the cytokine profiles of the MSCs compared to those of the control. Under inflammatory conditions, BCP 20/80 enhanced the secretion of FGF2, IL5 and CCL11, while BCP 60/40 enhanced CCL5 compared to that of the cytokine-stimulated controls (Figure 4).

### 2.3. BCP Altered MSC Gene Expression in Cocultures

In setup B, the paracrine interactions between the MPs and the MSCs in the presence of the BCP granules and/or cytokine stimulation were assessed by using a transwell coculture assay via the expressions of the same panel of genes analyzed in setup A. After 72 h, the coculture of the MPs with the monolayer MSCs enhanced the expressions of all the analyzed genes related to osteogenesis (*RUNX2* and *BMP2*), healing (*VEGF* and *IL10*), inflammation (*IL6* and *IL8*) and remodeling (*RANKL* and *OPG*) compared to those of the monolayer controls. In comparison, the coculture of the MPs with MSC + BCP 60/40 revealed the further upregulation of *BMP2, VEGF* and *OPG*, while the *IL10, IL6, IL8* and *RANKL* expressions remained unchanged. Under inflammatory conditions, BCP 60/40 remarkably enhanced the expressions of *BMP2, IL10, IL6* and *IL8* in the MSCs, while the remodeling genes (*RANKL* and *OPG*) were downregulated (Figure 5).

The coculture of the MPs with MSC + BCP 20/80 revealed the upregulation of *BMP2, VEGF* and *IL10* vs. the coculture of the MPs with the monolayer MSCs, while the inflammation (*IL6* and *IL8*) and remodeling genes (*RANKL* and *OPG*) remained unchanged. In contrast to BCP 60/40, the coculture of the MPs with MSC + BCP 20/80 under inflammatory conditions revealed the upregulation of the *IL10* gene only, whereas *IL6* and *IL8* were not upregulated in comparison to those in the control cocultures with cytokine stimulation (Figure 6).

The influences of the BCP-cultured MSCs on the gene expressions of the MPs were also assessed after 72 h of coculture. The genes commonly associated with the M1 (*IL1β, IL6* and *IL8*) and M2 MP phenotypes (*IL10* and *VEGF*) were evaluated. No significant differences in MP differentiation were observed when cocultured with either monolayer MSCs or BCP 20/80-cultured MSCs, regardless of cytokine stimulation. However, in the BCP 60/40-cultured MSCs, cytokine stimulation led to the upregulation of M1 macrophage markers (*IL1β, IL6* and *IL8*) (Appendix A). This suggests a predisposition towards the M1 MP subtype, providing further evidence of an acute inflammatory microenvironment, although no clear evidence of an MP phenotype “switch” was observed.

### 2.4. BCP Altered MSC Cytokine Profile in Cocultures

In setup B, the coculture with the MPs did not significantly alter the cytokine profiles of the MSCs based on the evaluated panel of cytokines. With regards to the BCP granules, the secretion of VEGF and CCL4 was enhanced in MSC + BCP 60/40, while IL5, IL7 and IL9 were enhanced in MSC + BCP 20/80 compared to those in the control cocultures. In the presence of cytokine stimulation, the secretion of GCSF, IL10 and IL9 was enhanced in MSC + BCP 60/40, while no increased secretion was observed in MSC + BCP 20/80 compared to that in the cytokine-stimulated control cocultures. In fact, the secretion of PDGFBB, IL7 and CCL5 was significantly reduced in stimulated MSC + BCP 20/80 (Figure 7).

## 3. Discussion

Tissue engineering strategies for bone regeneration frequently involve the use of MSCs seeded on biomaterials, e.g., alloplastic bone substitutes, used as carrier scaffolds. The objective of the present study was to assess whether BCP bone substitutes influence vitro MSC responses and MSC–MP interactions in an inflammatory microenvironment. The research question in a clinical context was whether, within the limitations of in vitro models, the presence of a bone substitute poses a risk for aggravating early cellular responses and, possibly, delaying healing at sites of active inflammation. MSCs were cultured in the presence of BCP granules (BCP 60/40 or 20/80) and cytokine stimulation (IL1β + TNFα) to mimic acute inflammation, either alone or in a coculture with MPs. Overall, our findings indicate that the BCP granules (a) significantly modulated MSC gene expressions, both in the presence and absence of inflammation; (b) did not significantly alter MSC cytokine secretion, regardless of inflammation; and (c) in the indirect coculture with the MPs, did not significantly alter MSC gene expressions or cytokine secretion, regardless of inflammation.

Emerging concepts suggest that the mechanisms of MSC bioactivity primarily involve the paracrine modulation of host responses rather than direct differentiation and tissue-specific cell replacement [6,27]. It has been proposed that MSCs exert their effects via interactions with resident immune cells in the early stages of wound healing. Moreover, the resulting paracrine secretions may continue to stimulate other immune cells over time and guide the healing process [28,29]. MSCs respond to inflammation by adjusting their immunoregulatory repertoire and by differentially modulating their gene expressions and cytokine profiles [30]. With regards to the inflammatory cytokines used herein, IL1β has been reported to prime MSCs towards anti-inflammatory and pro-trophic phenotypes in vitro, while TNFα triggers a more potent pro-inflammatory profile to instrument effective tissue repair [31]. The stimulation of MSCs with a combination of pro-inflammatory cytokines may lead to additional or synergistic effects, such as increased cytokine secretion [32]. While several studies have investigated the effects of inflammatory cytokines on MSCs [33], few studies have reported on the responses of MSCs in the presence of BCP bone substitutes [13].

In setup A, regardless of cytokine stimulation, the BCP granules enhanced the expressions of osteogenesis- (BMP2) and healing-related genes (*VEGF* and *IL10*) but suppressed those of inflammation-related genes (*IL6* and *IL8*) in the MSCs, suggesting positive effects of BCP in terms of pro-healing MSC activity. Similar trends in MSC responses were observed with BCP 60/40 and BCP 20/80. Surprisingly, the expression of *RUNX2* was not significantly altered by the BCP granules after 72 h, despite the strong upregulation of *BMP2*, which is reported to be an upstream regulator of *RUNX2* [10,11]. Similar results have been reported in previous studies regarding *RUNX2* expression by MSCs on BCP granules. One possible reason could be the relatively early time point (72 h) used in the present study, as previous studies analyzed *RUNX2* expression after 7 and 14 days [34,35]. Despite the changes in the gene expressions of the MSCs, the secretion of healing- and inflammation-related cytokines was not significantly altered by the presence of the BCPs, either BCP 60/40 or BCP 20/80. Moreover, while cytokine stimulation significantly altered the secretory profiles of the MSCs, the additional effect of the BCP granules under inflammatory conditions was minimal. Nevertheless, the presence of the BCP granules seemed to elicit a “pro-healing” response in the MSCs, at least at the gene level, in an inflammatory microenvironment. In context, a previous study has shown that the expressions of several pro-inflammatory genes were attenuated in MSCs cultured on BCP [36]. Together, these findings highlight the relevance of BCP and the microenvironment in MSC activity.

To better simulate the in vivo scenario, it is important to study MSC behaviors in the context of other cells. MSCs interact extensively with immune cells to drive the healing process, and recent evidence has shown the key role played by immune cells, particularly MPs, in the regulation of MSCs during bone regeneration [21,37,38]. In setup B in the present study, the coculture of the MPs with the MSCs in the presence of the BCP granules strongly promoted MSC gene expressions. Specifically, the presence of either BCP 60/40 or BCP 80/20 revealed enhanced MSC expressions of osteogenesis- (*BMP2*) and healing-related genes (*VEGF* and *IL10*), especially in the presence of inflammation. Interestingly, the expressions of inflammation- (*IL6* and *IL8*) and remodeling-related genes (*RANKL* and *OPG*) remained unchanged. In fact, in the presence of BCP 60/40, *RANKL* and *OPG* were downregulated in the MSCs under inflammatory conditions, suggesting that the BCP granules did not aggravate pro-inflammatory MSC responses when cocultured with the MPs.

Inflammation and/or other pathological stimuli lead naive macrophages (M0) to activate into either a classical, pro-inflammatory type (M1) or an alternative anti-inflammatory type (M2) [29]. Signaling molecules from non-activated MPs, particularly BMP2, have been implicated in MSC–MP crosstalk [39]. Our results reveal a sharp upregulation of *BMP2* gene expression in the MSCs when cocultured with the MPs. This expression was further upregulated in the presence of the BCP granules, together with a downregulation of *RANKL* expression. These observations suggest the commitment of MSCs towards an osteogenic phenotype in the presence of non-activated MPs, and they may reflect the physiological role of tissue-resident MPs in bone homeostasis. Further, the role of BCP in the coculture system is also of interest. While the coculture of the MPs with the control MSCs under inflammatory conditions promoted gene expression but not cytokine secretion, both gene expression and cytokine secretion were strongly promoted in the MSC–MP cocultures in the presence of BCP. Therefore, we hypothesize that the presence of BCP granules endorsed the cytokine stimulation of the MSCs, allowing for an accelerated protein translation.

Although two different commercial BCP bone substitutes were used in the present study (BCP 60/40 and 20/80), the objective herein was not to perform a biological comparison of the two biomaterials but rather to investigate whether a similar trend in MSC responses could be observed with BCP granules with different HA/β-TCP ratios. While the trends in MSC gene expressions and cytokine secretion were generally similar between the two BCPs, some differences were observed. For example, the MSCs showed remarkable differences in the secretion of GCSF in the presence of BCP 60/40 and BCP 20/80. GCSF has been shown to play distinct roles in normal state conditions, as well as in inflammatory conditions [40], and it has been shown to be produced in higher amounts by cytokine-stimulated MSCs [31,41], which could elucidate the differences observed herein. This was further demonstrated in the coculture setup, where the MSCs secreted higher levels of GCSF in the presence of BCP 60/40. Together, these findings highlight the impact of subtle biomaterial properties on the immunomodulatory responses of MSCs.

Some limitations of the present study must be acknowledged. Firstly, the results herein are based on MSCs derived from a single donor, and, therefore, the findings should be verified using multiple donors’ MSCs to exclude the effects of donor variation. Moreover, we differentiated MPs from a promonocytic cell line (U937) and not from primary peripheral blood monocytes. Although widely used as an economical and reliable in vitro model, U937-derived MPs may not accurately replicate the “plasticity” and/or responses of the M1/M2 phenotype in the context of other cells, i.e., in cocultures. For example, the increased secretion of anti-inflammatory cytokines observed in the BCP-cultured MSCs did not translate to altered gene expressions of the U937-derived MPs in the corresponding cocultures. A similar trend in LPS-treated BV2 cells has been observed when exposed to conditioned medium from cytokine-stimulated MSCs [41]. Cell-line-derived MPs are reported to differ from primary MPs in their cytokine profiles, which could also explain the differences in gene expression patterns [42]. Another limitation herein was the lack of functional assays to demonstrate MP activity, e.g., via direct culture on BCP, and to demonstrate MSC function, e.g., the suppression of T-cell proliferation, in order to support the gene and protein analyses. Finally, the role of other innate immune cells (particularly neutrophils) in MSC- and biomaterial-mediated healing should be investigated in future studies [43,44].

## 4. Materials and Methods

### 4.1. Cell Culture

The use of human cells and tissues was approved by the Regional Committees for Medical Research Ethics (REK) in Norway (2013-1248, REK sør-øst C). Primary human bone marrow MSCs from a healthy 10-year-old male donor were cultured in growth medium (GM) composed of Dulbecco’s modified Eagle’s medium (DMEM, Invitrogen, Carlsbad, CA, USA) supplemented with 1% penicillin/streptomycin (GE Healthcare, South Logan, UT, USA) and 10% fetal bovine serum (FBS; GE Healthcare). The details of MSC isolation and characterization via immunophenotyping and tri-lineage differentiation assays have been reported elsewhere [45]. Cells were sub-cultured (4000 cells/cm^2^) and expanded in humidified 5% CO_2_ at 37 °C; passage 2–4 cells were used in experiments.

MPs were derived from the human pro-monocytic U937 cell line (CRL-1593.2, ATCC, Rockville, MD, USA); cells were cultured in GM as described above. To induce differentiation into the MPs, U937 cells were stimulated with 50 ng/mL phorbol 12-myristate 13-acetate (PMA; Sigma-Aldrich, St. Louis, MO, USA) for 48 h [46]. Subsequently, the PMA-treated cells were washed in phosphate-buffered saline (PBS; Invitrogen), to remove the PMA along with nonadherent cells, and further maintained in GM. Cell growth and the morphology of the MSCs and MPs were regularly monitored under a phase-contrast microscope (Nikon Eclipse TS100, Tokyo, Japan).

### 4.2. BCP Bone Substitutes

Two different commercial BCP bone substitutes were used in this study: BoneCeramic^®^ (BC; Institut Straumann AG, Basel, Switzerland) porous granules (0.5–1 mm) with HA/β-TCP in a 60/40 ratio and Biomatlante MBCP+^®^ (BM; Biomatlante, Vigneux de Bretagne, France) micro-porous granules (0.5–1 mm) with HA/β-TCP in a 20/80 ratio. Both BCPs were supplied in sterile packaging and used under sterile conditions in the experiments. Both BCPs have previously been used to deliver MSCs in clinical studies of bone tissue engineering [47,48].

### 4.3. Experimental Setup

Two experimental setups were used in this study: setup A, where MSCs were seeded on BCP with and without cytokine stimulation, and setup B, where MSCs were seeded on BCP and cocultured with MPs with and without cytokine stimulation to simulate an inflammatory microenvironment. The experimental setups are summarized in Table 1.

### 4.4. Cell Seeding

The BCP 60/40 and BCP 20/80 granules (~100 mg per well) were separately loaded in 24-well tissue culture plates and pre-conditioned with GM overnight at 37 °C to promote cell attachment. Next, MSCs suspended in GM (150 × 10^3^ cells in 100 µL per well) were uniformly seeded on the granules and allowed to attach for 2 h. Subsequently, an additional 900 µL of GM (total 1 mL) was added and cultured for 72 h. Monolayer MSCs on the tissue culture plastic served as controls.

### 4.5. MSC–MP Coculture

In setup B, the cocultures of the MPs with the MSCs (1:4 MP:MSC) and BCP 60/40 or BCP 20/80 granules were set up via transwell assays using polyester membrane inserts with a 0.4 μm pore size (Corning, Lowell, MA, USA); transwell membranes allow cellular interactions without direct cell-to-cell contact. The MPs cocultured with the monolayer MSCs served as controls. The MSCs were seeded on the BCP granules in notched 24-well plates as described above. Separately, the U937 cells were seeded in transwell inserts and stimulated with PMA for 48 h to induce MP differentiation, and they were allowed to mature in GM for an additional 24 h. Thereafter, the inserts with adherent MPs were transferred to the notched wells with the MSCs, and the coculture was initiated in GM for an additional 72 h. In relevant groups, the culture media were supplemented with cytokines to simulate an inflammatory microenvironment.

### 4.6. Cytokine Stimulation

To simulate inflammation, the MSCs in setups A and B were stimulated with a combination of recombinant human IL1β (10 ng/mL) and TNFα (10 ng/mL) (both from R&D Systems, Minneapolis, MN, USA). Cytokines were added to GM in order to stimulate the MSCs for an additional 72 h, corresponding to the duration of the “acute inflammatory phase” in the in vivo wound-healing cascade. The expressions of genes and secretions of cytokines (proteins) were assessed in standard (unstimulated) and stimulated monolayer MSCs (control) and BCP-cultured MSCs.

### 4.7. Gene Expression Analysis

In setups A and B, the expressions of the genes associated with osteogenesis, healing, inflammation and remodeling (Appendix A) were assessed in the MSCs after 72 h. Gene expression was assessed via quantitative real-time polymerase chain reaction (qPCR) using TaqMan^®^ real-time PCR assays (Thermo Scientific). RNA extraction and cDNA synthesis were performed as previously described [45]. Briefly, total RNA was extracted using an RNA extraction kit (Maxwell, Promega, Madison, WI, USA), and cDNA was synthesized using a high-capacity cDNA reverse transcription kit (Applied Biosystems, Foster City, CA, USA), following the manufacturers’ protocols. qPCR was performed using a TaqMan Fast Universal PCR Master Mix with amplification in a StepOne Real-Time PCR System (both from Applied Biosystems), following the manufacturers’ protocols. The expressions of the genes of interest were normalized to that of the housekeeping gene glyceraldehyde 3-phosphate dehydrogenase (GAPDH). Data were analyzed by using the ΔΔ*Ct* method, and the results are presented as fold changes relative to the results of the control group (unstimulated monolayer MSCs).

### 4.8. Multiplex Cytokine Assay

In setups A and B, the concentrations of various cytokines (Appendix A) in the supernatant media of the MSCs were measured using a human cytokine 27-plex assay and the Bio-Plex^®^ 200 System (both from Bio-Rad Laboratories, CA, USA), according to the manufacturer’s instructions. Supernatant media from the MSCs in the different culture conditions were collected after 72 h for cytokine analyses. The total protein concentrations (µg/mL) in all samples were measured using a Pierce^®^ Bicinchoninic Acid Protein Assay (Thermo Scientific) according to the manufacturer’s instructions. As the total protein concentrations were significantly different between the groups, individual cytokine concentrations in the multiplex assay were normalized to the corresponding total protein (pg/µg) for each group.

### 4.9. Statistical Analysis

Statistical analyses were performed using Prism 9.0 software (GraphPad Software, San Diego, CA, USA). Data are presented as means (±SD) unless otherwise specified. Gene expression analyses are based on delta-CT values, and the results are presented as relative (log/non-linear) fold changes using scatter plots. All other linear data are presented as bar graphs. Normality testing was performed via the Shapiro–Wilk test. A one-way analysis of variance (ANOVA, followed by post hoc Tukey’s test for multiple comparisons) was applied, and *p* < 0.05 was considered statistically significant.

## 5. Conclusions

Overall, the findings herein indicate that, under both inflammatory and non-inflammatory conditions, the BCP granules significantly regulated the expressions of osteogenesis-, healing- and inflammation-related genes in the MSCs towards a pro-healing phenotype but had relatively little effect on the MSC secretory profiles. In the presence of the MPs (indirect coculture), BCP positively regulated both the gene expressions and cytokine secretion of the MSCs. Overall, similar trends in MSC responses were observed with BCP 60/40 and BCP 20/80. Thus, within the limitations of in vitro models, we postulate that the presence of a BCP bone substitute at the surgical site does not have a detrimental effect on MSC-mediated healing, even in the event of inflammation. Future studies using primary human immune cells may more accurately reveal the mechanisms of crosstalk with MSCs in the context of bone regeneration.

## Figures and Tables

**Figure 1 ijms-24-00438-f001:**
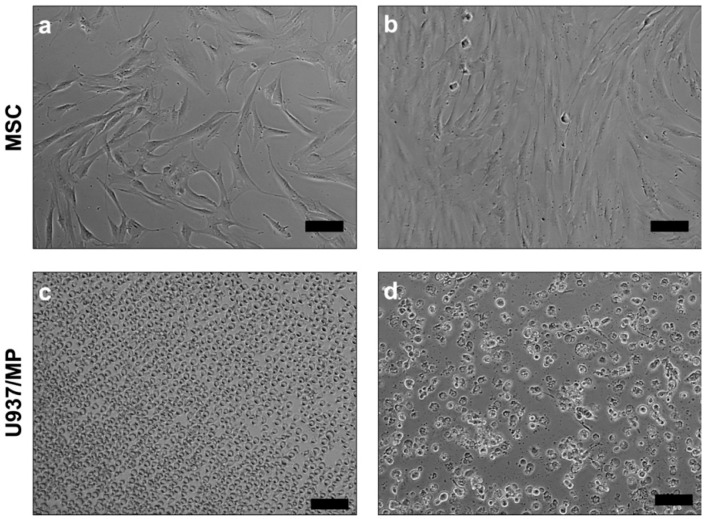
Representative phase contrast images of human bone-marrow-derived MSCs (**a**,**b**), U937 monocytes (**c**) and U937-derived macrophages (**d**); scale bars 100 µm.

**Figure 2 ijms-24-00438-f002:**
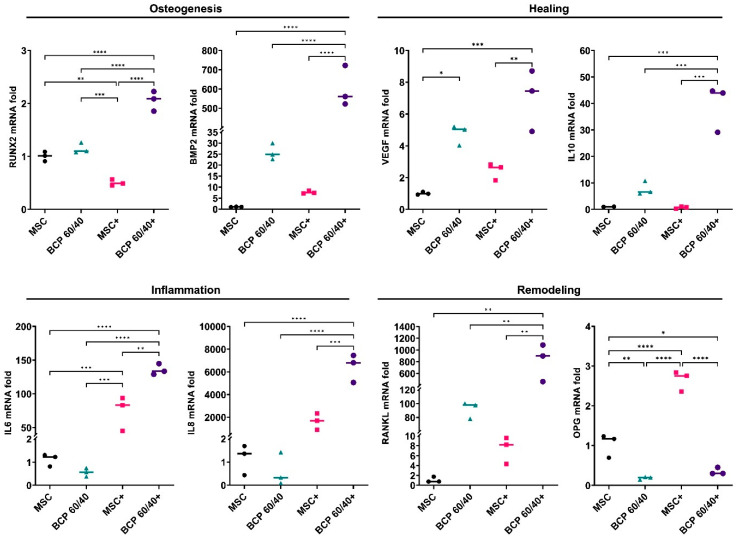
Relative mRNA expressions (fold changes) of osteogenesis-, healing-, inflammation- and remodeling-related genes in MSCs cultured as monolayers (control) or with BCP 60/40; + represents cytokine stimulation (n = 3). Statistical analyses are based on one-way ANOVA with Tukey’s multiple comparison tests on delta-Ct values; * *p* < 0.05; ** *p* < 0.001; *** *p* = 0.0001; **** *p* < 0.0001.

**Figure 3 ijms-24-00438-f003:**
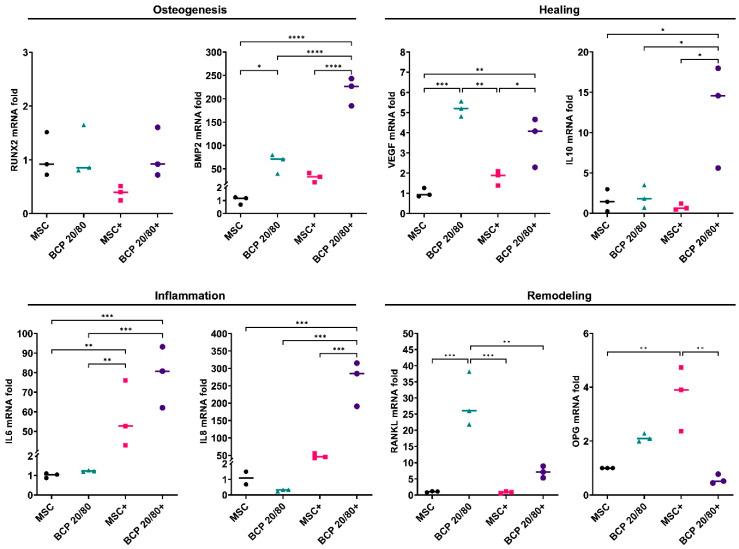
Relative mRNA expressions (fold changes) of osteogenesis-, healing-, inflammation- and remodeling-related genes in MSCs cultured as monolayers (control) or with BCP 20/80; + represents cytokine stimulation (n = 3). Statistical analyses are based on one-way ANOVA with Tukey’s multiple comparison tests on delta-Ct values; * *p* < 0.05; ** *p* < 0.001; *** *p* = 0.0001; **** *p* < 0.0001.

**Figure 4 ijms-24-00438-f004:**
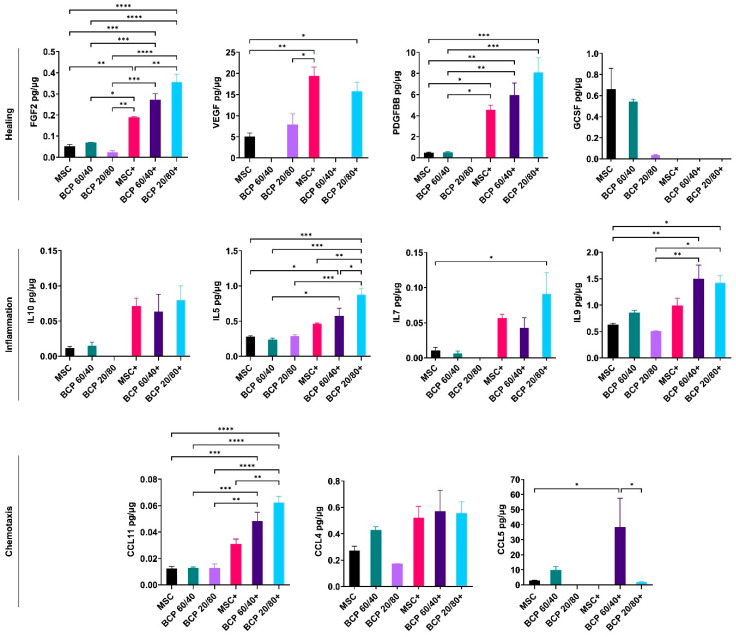
Multiplex cytokine assay of MSCs cultured with BCP granules (BCP 60/40 or BCP 20/80). Cytokines related to healing, inflammation and chemotaxis were measured after 72 h. + indicates cytokine stimulation. Concentration of each analyte (pg/mL) was normalized to total protein concentration of the conditioned media (µg/mL). Statistical analyses are based on one-way ANOVA with Tukey’s multiple comparison tests; * *p* < 0.05; ** *p* < 0.001; *** *p* = 0.0001; **** *p* < 0.0001.

**Figure 5 ijms-24-00438-f005:**
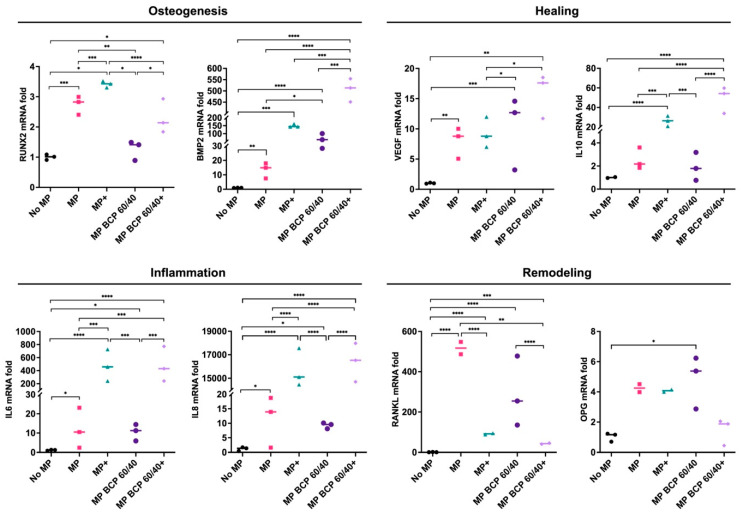
Relative mRNA expressions (fold changes) of osteogenesis-, healing-, inflammation- and remodeling-related genes in MSCs cultured with BCP 60/40 and/or MPs (n = 3). No MP, monolayer MSC control; MP, monolayer MSCs cocultured with MPs; + represents cytokine stimulation. Statistical analyses are based on one-way ANOVA with Tukey’s multiple comparison tests on delta-Ct values; * *p* < 0.05; ** *p* < 0.001; *** *p* = 0.0001; **** *p* < 0.0001.

**Figure 6 ijms-24-00438-f006:**
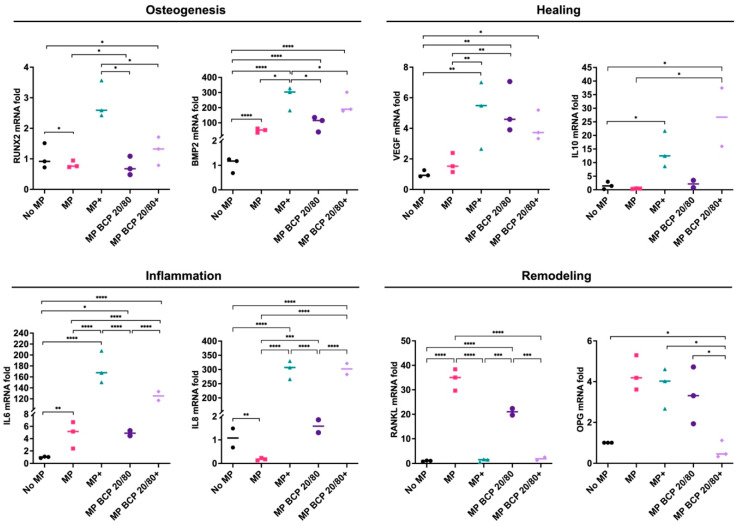
Relative mRNA expressions (fold changes) of osteogenesis-, healing-, inflammation- and remodeling-related genes in MSCs cultured with BCP 20/80 and/or MPs (n = 3). No MP, monolayer MSC control; MP, monolayer MSCs cocultured with MPs; + represents cytokine stimulation. Statistical analyses are based on one-way ANOVA with Tukey’s multiple comparison tests on delta-Ct values; * *p* < 0.05; ** *p* < 0.001; *** *p* = 0.0001; **** *p* < 0.0001.

**Figure 7 ijms-24-00438-f007:**
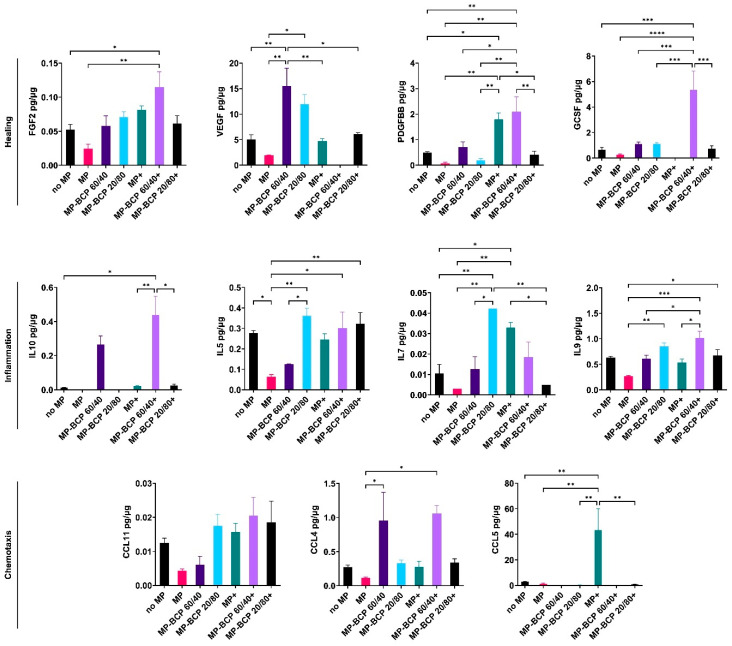
Multiplex cytokine assay of MPs co-cultured with MSCs and BCP granules (BCP 60/40 or BCP 20/80). Cytokines related to healing, inflammation and chemotaxis were measured after 72 h. + indicates cytokine stimulation. Concentration of each analyte (pg/mL) was normalized to total protein concentration of the conditioned media (µg/mL). Statistical analyses are based on one-way ANOVA with Tukey’s multiple comparison tests; * *p* < 0.05; ** *p* < 0.001; *** *p* = 0.0001; **** *p* < 0.0001.

**Table 1 ijms-24-00438-t001:** Summary of experimental setups and groups.

Setup, Group	
**A: MSC monoculture**	
MSC	MSC+
MSC/BCP 60/40 MSC/BCP 20/80	MSC/BCP 60/40+ MSC/BCP 20/80+
**B: MP-MSC coculture**	
MP-MSC	MP-MSC+
MP-MSC/BCP 60/40+ MP-MSC/BCP 20/80+	MP-MSC/BCP 60/40+ MP-MSC/BCP 20/80+

MSC, bone marrow mesenchymal stromal cell; BCP 60/40, BoneCeramic^®^; BCP 20/80, Biomatlante MBCP+^®^; +, cytokine stimulation; MP, U937-derived macrophage.

## Data Availability

All data are provided in either the figures or Appendix A; otherwise, they are available upon request.

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
