# Peer review of "Influence of Bone Substitutes on Mesenchymal Stromal Cells in an Inflammatory Microenvironment"

_ijms, 2022, doi:10.3390/ijms24010438_

Round 1

Reviewer 1 Report

The work conducted by Shanbhag et al. here reviewed is a piece of basic research that contains a lot of information on gene and protein expression via PCR and multiplex assays. The research design is appropriate, having employed two different BCPs, and the relevant control conditions. The results obtained might not be the best for an actual application in the clinical setting. However, the conclusions stated in the document are well supported by evidence and the major limitations of the work are listed at the end.

As suggestions, I have struggled to understand at first some abbreviations. As referred as BCP1 and BCP2 in the abstract, at the result sections appear as BC and BM, those abbreviations are not listed in the table of abbreviations as they are not molecules. It would be convenient to state once again at the beginning of the results section that these experiments have been carried out with two different commercial BCP, named them and make clear the abbreviations.

Also, there is no mention to the sterilization of BCPs. I assume that as they are commercial ones, they are handed sterile. If thats not the case, please comment on it.

Author Response

We thank the Editor and Reviewers for their valuable comments and feedback, which we believe have significantly improved the quality of our manuscript. Please find our responses to each of the reviewers’ comments below (in red) along with corresponding changes in the revised manuscript (highlighted with ‘track changes’). We hope that our responses are satisfactory and that the revised manuscript is now of an acceptable quality. We look forward to your decision. Thank you.

General comments

  1. All reviewers mention that some abbreviations, particularly related to the two BCP biomaterials (formerly BM and BC), were confusing. These have now been revised to represent the exact formulation of the biomaterials – BCP 60/40 and BCP 20/80, throughout the manuscript to avoid any confusion.

  1. The Results section has been reorganized to simplify the understanding and separate the gene level and protein level data for the two experimental setups in the study. Moreover, the data has been ‘grouped’ into sub-categories, e.g., osteogenesis-related genes, inflammation-related genes, etc., to simplify understanding.

  1. The Figures have been revised according to the Results and in their layout to make them more readable.

Reviewer 1

The work conducted by Shanbhag et al. here reviewed is a piece of basic research that contains a lot of information on gene and protein expression via PCR and multiplex assays. The research design is appropriate, having employed two different BCPs, and the relevant control conditions. The results obtained might not be the best for an actual application in the clinical setting. However, the conclusions stated in the document are well supported by evidence and the major limitations of the work are listed at the end.

We thank the Reviewer for this constructive feedback.

  1. As suggestions, I have struggled to understand at first some abbreviations. As referred as BCP1 and BCP2 in the abstract, at the result sections appear as BC and BM, those abbreviations are not listed in the table of abbreviations as they are not molecules. It would be convenient to state once again at the beginning of the results section that these experiments have been carried out with two different commercial BCP, named them and make clear the abbreviations.

We acknowledge that the previous abbreviations of the biomaterials used (BM and BC), may have been confusing. These have now been revised to represent the exact formulation of the biomaterials – BCP 60/40 and BCP 20/80, throughout the manuscript to avoid any confusion.

  1. Also, there is no mention to the sterilization of BCPs. I assume that as they are commercial ones, they are handed sterile. If thats not the case, please comment on it.

The BCPs used in the study were commercially available products intended for clinical use and therefore supplied sterile. They were under sterile conditions in all experiments. This has been clarified in a sentence in the Methods section of the revised manuscript.

Reviewer 2 Report

The manuscript entitled 'Influence of bone substitutes on mesenchymal stromal cells in an inflammatory microenvironment' by Shanbhag et al. investigated the effect of different culture conditions Mesenchymal stromal cells (MSCs) + bi-calcium phosphate (BCP) + inflammatory cytokine stimulation or coculture of MSCs with macrophages (MP) + BCP +  inflammatory cytokine stimulation on gene and protein expression. The gene expression of genes related to osteogenesis/healing (RUNX2, BMP2, VEGF), inflammation/immune modulation (IL6, IL8, IL10) and remodeling (RANK, OPG) were analyzed after 72h of culture. The objective of the study was to assess whether BCP bone substitutes influence MSC responses and MSC-MP interactions in an inflammatory microenvironment. The major finding of this manuscript is that BCP culture combined with an inflammatory microenvironment causes the strongest in vitro responses in MSC based on the enhanced expression of several genes and proteins related to wound healing and immunemodulation. The presence of BCP does not significantly alter the interactions between MSC and U937-derived MP in trans-well cocultures.

I have the following comments:

Major points:

1.    Why did the authors use just one primary human bone marrow source as indicated on p9 line 281. There could be differences between donors. To get objective results what happens in healthy MSC during the investigated culture conditions at least 3 donors with 3 replicates should be investigated. In addition, the donor was very young with 10 years could there be a difference with adult MSCs?

2.    The authors used BCP- cultures with BoneCeramic named BC-culture and Biomatlante named BM-culture this is very confusing especially on p6 lines 145-149. The authors should consider fusing data of the two materials as BCP-culture and lining out differences between the two BCPs in an extra figure.

3.    P2 line 94-97 the authors wrote: Expressions of genes related to osteogenesis/healing … There is no reference indicating were this knowledge comes from and it is not further explained in the introduction why these genes were investigated.

Minor points:

1.    P5 line 144 ..MSC was studies… should be corrected

2.    Could the experiments be performed without trans-well to have direct cell-cell contact?

Author Response

We thank the Editor and Reviewers for their valuable comments and feedback, which we believe have significantly improved the quality of our manuscript. Please find our responses to each of the reviewers’ comments below (in red) along with corresponding changes in the revised manuscript (highlighted with ‘track changes’). We hope that our responses are satisfactory and that the revised manuscript is now of an acceptable quality. We look forward to your decision. Thank you.

General comments

  1. All reviewers mention that some abbreviations, particularly related to the two BCP biomaterials (formerly BM and BC), were confusing. These have now been revised to represent the exact formulation of the biomaterials – BCP 60/40 and BCP 20/80, throughout the manuscript to avoid any confusion.

  1. The Results section has been reorganized to simplify the understanding and separate the gene level and protein level data for the two experimental setups in the study. Moreover, the data has been ‘grouped’ into sub-categories, e.g., osteogenesis-related genes, inflammation-related genes, etc., to simplify understanding.

  1. The Figures have been revised according to the Results and in their layout to make them more readable.

Reviewer 2

The manuscript entitled 'Influence of bone substitutes on mesenchymal stromal cells in an inflammatory microenvironment' by Shanbhag et al. investigated the effect of different culture conditions Mesenchymal stromal cells (MSCs) + bi-calcium phosphate (BCP) + inflammatory cytokine stimulation or coculture of MSCs with macrophages (MP) + BCP +  inflammatory cytokine stimulation on gene and protein expression. The gene expression of genes related to osteogenesis/healing (RUNX2, BMP2, VEGF), inflammation/immune modulation (IL6, IL8, IL10) and remodeling (RANK, OPG) were analyzed after 72h of culture. The objective of the study was to assess whether BCP bone substitutes influence MSC responses and MSC-MP interactions in an inflammatory microenvironment. The major finding of this manuscript is that BCP culture combined with an inflammatory microenvironment causes the strongest in vitro responses in MSC based on the enhanced expression of several genes and proteins related to wound healing and immune modulation. The presence of BCP does not significantly alter the interactions between MSC and U937-derived MP in trans-well cocultures. 

We thank the Reviewer for this constructive feedback.

I have the following comments:

Major points:

  1. Why did the authors use just one primary human bone marrow source as indicated on p9 line 281. There could be differences between donors. To get objective results what happens in healthy MSC during the investigated culture conditions at least 3 donors with 3 replicates should be investigated. In addition, the donor was very young with 10 years could there be a difference with adult MSCs?

The MSCs used in the present study (and several previous studies in our research group) were obtained from an existing Biobank at our Institute. This Biobank primarily contains bone marrow MSCs from systemically healthy individuals aged 8-12 y undergoing surgery involving bone marrow harvest. We have previously published several studies using MSCs from this Biobank, also in this journal (PMID: 34948255). In preliminary experiments, no significant differences have been observed between cells from young (8-12 y) and older donors (18-35 y) based on basic MSC properties of proliferation and multilineage differentiation. Nevertheless, we acknowledge that using cells from only one single donor is a limitation of the study and the findings should be verified using multiple donors’ MSC to exclude the effects of donor variation We have now included a sentence in the Discussion to reflect this.

  1. The authors used BCP- cultures with BoneCeramic named BC-culture and Biomatlante named BM-culture this is very confusing especially on p6 lines 145-149. The authors should consider fusing data of the two materials as BCP-culture and lining out differences between the two BCPs in an extra figure.

We acknowledge that the previous abbreviations of the biomaterials used (BM and BC), may have been confusing. These have now been revised to represent the exact formulation of the biomaterials – BCP 60/40 and BCP 20/80, throughout the manuscript to avoid any confusion. Although the general trends of MSC responses were similar on the two biomaterials, some differences were observed (as reported in the manuscript), and therefore the data has been presented separately for the two BCP materials.

  1. P2 line 94-97 the authors wrote: Expressions of genes related to osteogenesis/healing … There is no reference indicating where this knowledge comes from and it is not further explained in the introduction why these genes were investigated. 

We thank the Reviewer for this valuable suggestion. We have now introduced the genes/ proteins of interest in the Introduction along with their function and clinical relevance. We have also categorized the genes/proteins into groups to simplify the understanding.

Minor points:

  1. P5 line 144 ..MSC was studies… should be corrected

This has now been corrected.

  1. Could the experiments be performed without trans-well to have direct cell-cell contact?

Indeed, the coculture experiments could be performed via direct cell-cell contact. However, in the present study we were mainly interested in the paracrine effects of MP on MSC, and vice versa, and therefore an indirect coculture model was selected. This has now been clarified in the revised manuscript.  

Reviewer 3 Report

Subject of manuscript no. 2075657 is interesting, as are the results obtained. However, the manuscript in its current form is unsuitable for publication and requires major corrections. In particular, the description of the results needs improvement.

Introduction

The authors should briefly describe the role of the studied genes / proteins in the wound healing process. The authors can classify genes into groups (as in Supplementary Table 1) and briefly describe their meaning.

Materials and Methods

In Table 1, the abbreviations used by the authors in the description of the results and in Figures 1-7 should be put in parentheses (brackets).

Lines 343-344: Information on the isolation of mRNA from cells should be briefly described in this chapter, even despite the citation

Supplementary table 1- The explanation of PPARG and SOX9 abbreviations in the table description should be deleted as these genes are not included in the table.

Results

Most of the issues concern this part of the work. The results seem inconsistent. The authors have analyzed a wide panel of growth factors involved in bone osteogenesis, remodeling and inflammation but do not fully explain to the reader what their role in these processes is. The gene expression profile should be supplemented with a protein profile (Western-Blot analysis).

 Moreover, the description of the results is imprecise and sometimes unclear.

There are discrepancies between the description and the graphic form of the results.

Line 103-104: RUNX2 was upregulated only in BC-cultured MSC (Figures 2 and 3). What about BM- cultured MSC ?? -  the RUNX2 chart (Figure 3) shows the statistical significance p <0.05 (*).

OPG mRNA expression data has not been described.

Line 131- should be FGF2

Line 130-132- The combination of BCP culture and cytokine stimulation revealed the highest secretion of healing- (FGF, PDGF-BB) and inflammation/immune modulation-related cytokines (IL10, IL5, IL7, IL9, IL17, CCL11, CCL4, IFNγ); however, there are no statistical significances for IL10 and IL17 in the graphs???

There is no data for CCL4

Figure 4- Abbreviations above the graphs should be corrected because they contain errors:

CCL1- should be CCL11- as described in the text

IFNG- should be IFN gamma (γ)

MIP1B- should be Beta (β)

Moreover, the chart of the concentration of CCL4 protein is missing.

The legend in Supplementary Figure1 is incomprehensible

Lines 167-169: After 72 h, coculture with MP revealed  increased secretion of VEGF, IL10 and CCL4 in BC-cultured MSC, and of VEGF, IL5, IL7, IL9 and CCL11 in BM-cultured MSC (Figure 7). However, there are no statistical significances in the plots of IL10, CCL11 expression in BC and BM-cultured MSC respectively.

CCL4- there is no data provided

Lines 169-172: Under cytokine-stimulation, coculture  with MP revealed marked increase in secretion of FGF2, PDGF-BB, GCSF, IL10, IL9, IL17,  CCL11 and CCL4 in BC-cultured MSC, while only CCL11 secretion was elevated in BM- cultured MSC (Figure 7). The same problem goes here. There are no statistically significant differences marked in the graphs of IL 17 and CCL 11, so it is difficult to talk about an increase in secretion in this case. What about the IFNγ and MIP1β concentrations?

The charts are too small and therefore hard to read.

Author Response

We thank the Editor and Reviewers for their valuable comments and feedback, which we believe have significantly improved the quality of our manuscript. Please find our responses to each of the reviewers’ comments below (in red) along with corresponding changes in the revised manuscript (highlighted with ‘track changes’). We hope that our responses are satisfactory and that the revised manuscript is now of an acceptable quality. We look forward to your decision. Thank you.

General comments

  1. All reviewers mention that some abbreviations, particularly related to the two BCP biomaterials (formerly BM and BC), were confusing. These have now been revised to represent the exact formulation of the biomaterials – BCP 60/40 and BCP 20/80, throughout the manuscript to avoid any confusion.

  1. The Results section has been reorganized to simplify the understanding and separate the gene level and protein level data for the two experimental setups in the study. Moreover, the data has been ‘grouped’ into sub-categories, e.g., osteogenesis-related genes, inflammation-related genes, etc., to simplify understanding.

  1. The Figures have been revised according to the Results and in their layout to make them more readable.

Reviewer 3

Subject of manuscript no. 2075657 is interesting, as are the results obtained. However, the manuscript in its current form is unsuitable for publication and requires major corrections. In particular, the description of the results needs improvement.

We thank the Reviewer for this constructive feedback. We are especially grateful for the careful observation of reporting errors and inconsistency between the data shown in the figures and in the text. We have carefully examined the revised manuscript for inconsistencies and confirm that all data reported is now consistent with the figures in the revised manuscript.

Introduction

The authors should briefly describe the role of the studied genes / proteins in the wound healing process. The authors can classify genes into groups (as in Supplementary Table 1) and briefly describe their meaning.

We thank the Reviewer for this valuable suggestion. We have now introduced the genes/ proteins of interest in the Introduction along with their function and clinical relevance. We have also categorized the genes/proteins into groups to simplify the understanding.

Materials and Methods

In Table 1, the abbreviations used by the authors in the description of the results and in Figures 1-7 should be put in parentheses (brackets).

Lines 343-344: Information on the isolation of mRNA from cells should be briefly described in this chapter, even despite the citation

Supplementary table 1- The explanation of PPARG and SOX9 abbreviations in the table description should be deleted as these genes are not included in the table.

All formatting errors have been corrected and missing information has been added in the revised manuscript.

Results

Most of the issues concern this part of the work. The results seem inconsistent. The authors have analyzed a wide panel of growth factors involved in bone osteogenesis, remodeling and inflammation but do not fully explain to the reader what their role in these processes is.

The Results section has been completely reorganized to simplify the understanding and separate the gene level and protein level data for the two experimental setups in the study. Moreover, the data has been ‘grouped’ into sub-categories, e.g., osteogenesis-related genes, inflammation-related genes, etc., to simplify understanding.

The gene expression profile should be supplemented with a protein profile (Western-Blot analysis).

The gene expression data has been supplemented with a protein profile, namely ELISA, for the most strongly modulated gene – BMP2. This data has been included in the Supplementary file (Suppl. Fig. S2)

There are discrepancies between the description and the graphic form of the results. Line 103-104: RUNX2 was upregulated only in BC-cultured MSC (Figures 2 and 3). What about BM-cultured MSC? - the RUNX2 chart (Figure 3) shows statistical significance p <0.05 (*).

OPG mRNA expression data has not been described.

Line 131- should be FGF2

Line 130-132- The combination of BCP culture and cytokine stimulation revealed the highest secretion of healing- (FGF, PDGF-BB) and inflammation/immune modulation-related cytokines (IL10, IL5, IL7, IL9, IL17, CCL11, CCL4, IFNγ); however, there are no statistical significances for IL10 and IL17 in the graphs?

There is no data for CCL4

Figure 4- Abbreviations above the graphs should be corrected because they contain errors: CCL1 should be CCL11 as described in the text, IFNG- should be IFN gamma (γ), MIP1B- should be Beta (β). Moreover, the chart of the concentration of CCL4 protein is missing.

The legend in Supplementary Figure1 is incomprehensible

Lines 167-169: After 72 h, coculture with MP revealed  increased secretion of VEGF, IL10 and CCL4 in BC-cultured MSC, and of VEGF, IL5, IL7, IL9 and CCL11 in BM-cultured MSC (Figure 7). However, there are no statistical significances in the plots of IL10, CCL11 expression in BC and BM-cultured MSC respectively. 

CCL4 – there is no data provided

Lines 169-172: Under cytokine-stimulation, coculture  with MP revealed marked increase in secretion of FGF2, PDGF-BB, GCSF, IL10, IL9, IL17,  CCL11 and CCL4 in BC-cultured MSC, while only CCL11 secretion was elevated in BM- cultured MSC (Figure 7). The same problem goes here. There are no statistically significant differences marked in the graphs of IL 17 and CCL 11, so it is difficult to talk about an increase in secretion in this case. What about the IFNγ and MIP1β concentrations?

We are grateful for the careful observation of reporting errors and inconsistency between the data shown in the figures and in the text. We have carefully examined the revised manuscript for inconsistencies and confirm that all data reported, including statistical significance, is now consistent with the figures in the revised manuscript.

The charts are too small and therefore hard to read.

The Figures have been revised according to the Results and in their layout to make them more readable.

Round 2

Reviewer 2 Report

The authors rivised their mauscript entirely and answered all of my questions well. Beside the limitation of the model (just one source of  MSCs) the manuscript showed that that the presence of a BCP bone substitute at the surgical site may not have a detrimental effect on MSC-mediated wound healing.

Reviewer 3 Report

The manuscript was revised in accordance with all comments from the reviewer